# UNITER: Learning UNiversal Image-TExt Representations

## Abstract

Joint image-text embedding is the bedrock for most Vision-and-Language (V+L) tasks, where multimodality inputs are jointly processed for visual and textual understanding. In this paper, we introduce UNITER, a UNiversal Image-TExt Representation, learned through large-scale pre-training over four image-text datasets (COCO, Visual Genome, Conceptual Captions, and SBU Captions), which can power heterogeneous downstream V+L tasks with joint multimodal embeddings. We design three pre-training tasks: Masked Language Modeling (MLM), Image-Text Matching (ITM), and Masked Region Modeling (MRM, with three variants). Different from concurrent work on multimodal pre-training that apply joint random masking to both modalities, we use conditioned masking on pre-training tasks (i.e., masked language/region modeling is conditioned on full observation of image/text). Comprehensive analysis shows that conditioned masking yields better performance than unconditioned masking. We also conduct a thorough ablation study to find an optimal setting for the combination of pre-training tasks. Extensive experiments show that UNITER achieves new state of the art across six V+L tasks (over nine datasets), including Visual Question Answering, Image-Text Retrieval, Referring Expression Comprehension, Visual Commonsense Reasoning, Visual Entailment, and NLVR$^2$.

## 1 Introduction

Most Vision-and-Language tasks rely on joint multimodel embeddings to bridge the semantic gap between visual and textual clues in images and text, although such representations are usually tailored for specific tasks. For example, MCB (Fukui et al., 2017), BAN (Kim et al., 2018), DFAF (Gao et al., 2019) proposed advanced multimodal fusion methods for Visual Question Answering (VQA) (Antol et al., 2015). SCAN (Lee et al., 2018) and MAttNet (Yu et al., 2018) studied learning latent alignment between words and image regions for Image-Text Retrieval (Wang et al., 2016) and Referring Expression Comprehension (Kazemzadeh et al., 2014) tasks. While each of these proposed models has pushed the state of the art on respective benchmarks, their architectures are diverse and the learned representations are highly task-specific, preventing them from being generalized to other tasks. This raises a million-dollar question: can we learn a universal image-text representation for all V+L tasks?

To answer this question, we introduce **UN**iversal **I**mage-**TE**xt **R**epresentations (**UNITER**), a large-scale pre-trained model for multimodal embedding. We adopt Transformer (Vaswani et al., 2017) as the core of our model, to leverage its elegant self-attention mechanism designed for learning contextualized representations. Inspired by BERT (Devlin et al., 2019), which has successfully applied Transformer to NLP tasks through large-scale language modeling, we pre-train UNITER through three pre-training tasks: ($i$) Masked Language Modeling (MLM) *conditioned on image*; ($ii$) Masked Region Modeling (MRM) *conditioned on text*; and ($iii$) joint Image-Text Matching (ITM). To further investigate the effectiveness of MRM, we propose three MRM variants: ($i$) Masked Region Classification (MRC); ($ii$) Masked Region Feature Regression (MRFR); and ($iii$) Masked Region Classification with KL-divergence (MRC-kl).

As shown in Figure 1, UNITER first encodes image regions (visual features and bounding box features) and textual words (tokens and positions) into a common embedding space with Image Embedder and Text Embedder, then applies a Transformer module to learn generalizable contex-

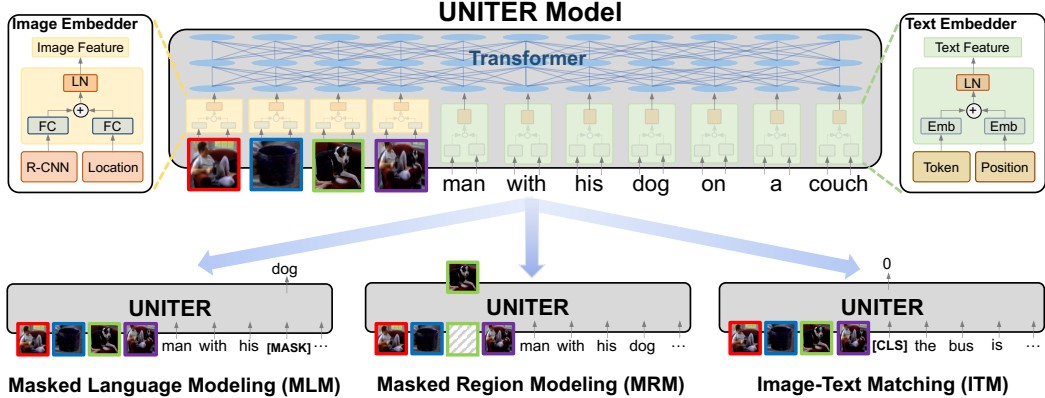

Figure 1: Overview of the proposed UNITER model (best viewed in color), consisting of an Image Embedder, a Text Embedder and a multi-layer self-attention Transformer, learned through three pre-training tasks.

tualized embeddings for each region and word through aforementioned pre-training tasks. Compared with LXMERT (Tan & Bansal, 2019) and ViLBERT (Lu et al., 2019) that use two streams (one Transformer for each modality), our UNITER model can learn joint contextualized representations for image regions and textual words through a single Transformer. Besides, our masked language/region modeling is conditioned on full observation of image/text, different from other concurrent pre-trained models that apply joint random masking to both modalities. We show that the conditional masking strategy can successfully ease the missing-alignment between images and text, and obtain better joint embeddings for downstream tasks. Detailed ablation study also demonstrates that the combination of MLM+ITM+MRC-kl+MRFR yields the best pre-training performance.

To demonstrate the power of UNITER, we evaluate on six V+L tasks across nine datasets, including: ($i$) VQA; ($ii$) Visual Commonsense Reasoning (VCR) (Zellers et al., 2019); ($iii$) NLVR$^2$ (Suhr et al., 2019); ($iv$) Visual Entailment (Xie et al., 2019); ($v$) Image-Text Retrieval (including zero-shot setting) (Lee et al., 2018); and ($vi$) Referring Expression Comprehension. Our UNITER model is trained on a large-scale V+L dataset composed of four subsets: ($i$) COCO (Lin et al., 2014); ($ii$) Visual Genome (VG) (Krishna et al., 2017); ($iii$) Conceptual Captions (CC) (Sharma et al., 2018); and ($iv$) SBU Captions (Ordonez et al., 2011). Experiments show that UNITER achieves new state of the art with significant performance boost across all six downstream tasks. Moreover, training on additional CC and SBU data (containing unseen images/text in downstream tasks) further boosts model performance over training on COCO and VG only.

Our contributions can be summarized as follows: ($i$) We introduce UNITER, a powerful UNiversal Image-TExt Representations for Vision-and-Language tasks. ($ii$) We achieve new state of the art (SOTA) on multiple V+L benchmarks, outperforming existing SOTA and concurrent multimodal pre-training methods by a large margin. ($iii$) We present extensive experiments and analysis to provide useful insights on the effectiveness of each pre-training task/dataset for multimodal encoder training.

## 2 RELATED WORK

Self-supervised learning utilizes original data as its own source of supervision, which has been applied to many Computer Vision tasks, such as image colorization (Zhang et al., 2016), solving jigsaw puzzles (Noroozi & Favaro, 2016; Trinh et al., 2019), inpainting (Pathak et al., 2016), rotation prediction (Gidaris et al., 2018), and relative location prediction (Doersch et al., 2015). Recently, pre-trained language models such as ELMo (Peters et al., 2018), BERT (Devlin et al., 2019), GPT2 (Radford et al., 2019), and XLNet (Yang et al., 2019) have shown great advances for NLP tasks. There are two keys to their success: effective pre-training tasks over large language corpus, and the use of Transformer (Vaswani et al., 2017) for learning contextualized text representations.

More recently, there has been some concurrent work on self-supervised learning for multimodal tasks, by pre-training on large-scale image/video and text pairs, then finetuning on downstream tasks. For example, VideoBERT (Sun et al., 2019) applied BERT to learn a bidirectional joint distribution over quantized video frame features and linguistic tokens from video-text pairs. ViLBERT (Lu

et al., 2019) and LXMERT (Tan & Bansal, 2019) introduced the two-stream architecture, where two Transformers are applied to images and text independently, which will be fused by a third Transformer in a later stage. On the other hand, VisualBERT (Li et al., 2019b), Unicoder-VL (Li et al., 2019a), VL-BERT (Su et al., 2019) and B2T2 (Alberti et al., 2019) proposed the single-stream architecture, where a single Transformer is applied to both image and text. Specifically, LXMERT model was pre-trained with downstream tasks such as VQA (Antol et al., 2015) and GQA (Hudson & Manning, 2019), while the others were pre-trained on image-text pairs only. Our UNITER model belongs to the second family. One key difference between UNITER and the other methods is the masking approach on pre-training tasks. Instead of randomly masking both image regions and sentence words, we use conditional masking, i.e., masking only one modality while keeping the other untainted. In addition, we examine the best combination of pre-training tasks through a thorough ablation study on the effects of each pre-training task and dataset on downstream tasks.

Another related work is DDAF (Gao et al., 2019), which proposed a novel architecture of inter-modality and intra-modality attention modules to learn the latent alignment between two modalities for VQA. Compared with Gao et al. (2019), UNITER learns a relatively more generic V+L representation via pre-training.

## 3 UNIVERSAL IMAGE-TEXT REPRESENTATIONS

In this section, we first introduce the model architecture of UNITER (Section 3.1), then describe the designed pre-training tasks and V+L datasets used for pre-training (Section 3.2 and 3.3).

### 3.1 MODEL OVERVIEW

The model architecture of UNITER is illustrated in Figure 1. Given a pair of image and sentence, UNITER takes the visual regions of the image and textual tokens of the sentence as the input. We design an Image Embedder and a Text Embedder to extract their respective embeddings. These embeddings are then feed it into a multi-layer self-attention Transformer to learn a cross-modality contextualized embedding between visual regions and textual tokens. Note that the self-attention mechanism in Transformer is order-less, thus it is necessary to explicitly encode positions/locations of tokens/regions as additional inputs.

Specifically, in *Image Embedder*, we first use Faster R-CNN[1] to extract the visual features (pooled ROI features) for each region. We also encode the location features for each region via a 7-dimensional vector[2]. Both visual and location features are then fed through a fully-connected (FC) layer, to be projected into the same embedding space. The final visual embedding for each region is obtained by summing up the two FC outputs and then passing through a layer normalization (LN) layer. For *Text Embedder*, we follow BERT (Devlin et al., 2019) and tokenize the input sentence into WordPieces (Wu et al., 2016). The final representation for each sub-word token[3] is obtained via summing up its word embedding and position embedding, followed by another LN layer[4].

We introduce three main tasks to pre-train our model: Masked Language Modeling *conditioned on image regions* (MLM), Masked Region Modeling *conditioned on input text* (with three variants) (MRM), and Image-Text Matching (ITM). As shown in Figure 1, our MRM and MLM are in analogy to BERT, where we randomly mask some words or regions from the input and learn to recover the words or regions as the output of Transformer. Specifically, word masking is realized by replacing the token with a special token [MASK], and region masking is implemented by replacing the visual feature vector with all zeros. Note that each time we only mask one modality while keeping the other modality intact, instead of randomly masking both modalities like ViLBERT and LXMERT. This prevents potential miss-alignment when a masked region happens to be described by a masked word. Empirically, we show that with conditional masking, our model is able to learn better embeddings

---

[1]Our Faster R-CNN was pre-trained on Visual Genome object+attribute data (Anderson et al., 2018).

[2]$[x_1, y_1, x_2, y_2, w, h, w * h]$ (normalized top/left/bottom/right coordinates, width, height, and area.)

[3]We use word/sub-word and token interchangeably throughout the rest of the paper.

[4]We also use a special modality embedding to help the model distinguish between textual and visual input, which is similar to the 'segment embedding' in BERT. This embedding is also summed before the LN layer in each embedder. For simplicity, this modality embedding is omitted in Figure 1.

(in Section 4.2). Lastly, we also learn an instance-level alignment (rather than token/region-level) between the whole image and the sentence via ITM. During training, we sample both positive and negative image-sentence pairs and learn their matching scores.

To pre-train UNITER with the aforementioned different tasks, we randomly sample one pre-training task for each mini-batch and train on only one objective per SGD update.

## 3.2 Pre-training Tasks

**Masked Language Modeling (MLM)** We denote the image regions as $\mathbf{v} = \{v_1, ..., v_K\}$, the input words as $\mathbf{w} = \{w_1, ..., w_T\}$, and the mask indices as $\mathbf{m} \in \mathbb{N}^M$. [5] In MLM, we randomly mask out the input words with probability of 15%, and replace the masked ones $\mathbf{w_m}$ with special token [MASK] [6]. The goal is to predict these masked words based on the observation of their surrounding words $\mathbf{w}_{\backslash m}$ and all image regions $\mathbf{v}$, by minimizing the negative log-likelihood:

$$\mathcal{L}_{\text{MLM}}(\theta) = -E_{(\mathbf{w},\mathbf{v})\sim D} \log P_\theta(\mathbf{w_m}|\mathbf{w}_{\backslash \mathbf{m}}, \mathbf{v}). \tag{1}$$

where $\theta$ is the trainable parameters. Each pair $(\mathbf{w}, \mathbf{v})$ is sampled from the whole training set $D$.

**Image-Text Matching (ITM)** In ITM, an additional special token [CLS] is fed into our model, which indicates the fused representation of both modalities. The inputs to ITM are a sentence and a set of image regions, and the output is a binary label (0 for negative match, and 1 for positive match). We extract the representation of [CLS] token as the joint representation of the input text and image, then fed into a fully connected layer and a sigmoid function to predict a score between 0 and 1. We denote the output score as $s_\theta(\mathbf{w}, \mathbf{v})$. The ITM supervision is over the [CLS] token. [7] During training, we sample a positive or negative pair $(\mathbf{w}, \mathbf{v})$ from the dataset $D$ at each step. The negative pair is created by replacing the image or text in a paired sample with a randomly-selected one from other samples. We denote the label as $y \in \{0, 1\}$, indicating if the sampled pair is a match. Then we apply a binary cross-entropy loss for optimization:

$$\mathcal{L}_{\text{ITM}}(\theta) = -E_{(\mathbf{w},\mathbf{v})\sim D}[y \log s_\theta(\mathbf{v}, \mathbf{w}) + (1 - y) \log(1 - s_\theta(\mathbf{v}, \mathbf{w}))]). \tag{2}$$

**Masked Region Modeling (MRM)** Similar to MLM, we also sample image regions and mask their visual features with a probability of 15%. The model is trained to reconstruct the masked regions $\mathbf{v_m}$ given the remaining regions $\mathbf{v}_{\backslash \mathbf{m}}$ and all the words $\mathbf{w}$. The visual features $v_m$ of the masked region are replaced by zeros. Unlike textual tokens that are represented as discrete labels, visual features are high-dimensional and continuous, thus cannot be supervised via class likelihood. Instead, we propose three variants for Masked Region Modeling, which share the same objective base:

$$\mathcal{L}_{\text{MRM}}(\theta) = E_{(\mathbf{w},\mathbf{v})\sim D} f_\theta(\mathbf{v_m}|\mathbf{v}_{\backslash \mathbf{m}}, \mathbf{w}). \tag{3}$$

1) **Masked Region Feature Regression (MRFR)** MRFR learns to regress the Transformer output of each masked region $\mathbf{v_m}^{(i)}$ to its visual features. Specifically, we apply an FC layer to convert its Transformer output into a vector $h_\theta(\mathbf{v_m}^{(i)})$ of same dimension as the input ROI pooled feature $r(\mathbf{v_m}^{(i)})$. Then we apply L2 regression between the two: $f_\theta(\mathbf{v_m}|\mathbf{v}_{\backslash \mathbf{m}}, \mathbf{w}) = \sum_{i=1}^{M} \|h_\theta(\mathbf{v_m}^{(i)}) - r(\mathbf{v_m}^{(i)})\|_2^2$.

2) **Masked Region Classification (MRC)** MRC learns to predict the object semantic class for each masked region. We first feed the Transformer output of the masked region $\mathbf{v_m}^{(i)}$ into an FC layer to predict the scores of $K$ object classes, which further goes through a softmax function to be transformed into a normalized distribution $g_\theta(\mathbf{v_m}^{(i)}) \in \mathbb{R}^K$. Note that there is no ground-truth label, as the object categories are not provided. Thus, we use the object detection output from Faster R-CNN, and take the detected object category (with the highest confidence score) as the label of the masked region, which will be converted into a one-hot vector $c(\mathbf{v_m}^{(i)}) \in \mathbb{R}^K$. The final objective minimizes the cross-entropy (CE) loss: $f_\theta(\mathbf{v_m}|\mathbf{v}_{\backslash \mathbf{m}}, \mathbf{w}) = \sum_{i=1}^{M} \text{CE}(c(\mathbf{v_m}^{(i)}), g_\theta(\mathbf{v_m}^{(i)}))$.

---

[5] $\mathbb{N}$ is the natural numbers, $M$ is the number of masked tokens, and $\mathbf{m}$ is the set of masked indices.

[6] Following BERT, we decompose this 15% into 10% random word, 10% unchanged, and 80% [MASK].

[7] The supervision over the [CLS] token in pretraining also alleviates the input mismatch problem between pretraining tasks and downstream finetuning tasks, since most of the downstream tasks take the representation of [CLS] token as the joint representation.

| Split | In-domain | | Out-of-domain | |
| --- | --- | --- | --- | --- |
| | COCO Captions | VG Dense Captions | Conceptual Captions | SBU Captions |
| train | 533K (106K) | 5.06M (101K) | 3.0M (3.0M) | 990K (990K) |
| val | 25K (5K) | 106K (2.1K) | 14K (14K) | 10K (10K) |

Table 1: Statistics on datasets used for pre-training. Each cell shows #image-text pairs (#images).

| | Task | Datasets | Image Src. | #Images | #Text | Metric |
| --- | --- | --- | --- | --- | --- | --- |
| 1 | VQA | VQA | COCO | 204K | 1.1M | VQA-score |
| 2 | VCR | VCR | Movie Clips | 110K | 290K | Accuracy |
| 3 | NLVR$^2$ | NLVR$^2$ | Web Crawled | 214K | 107K | Accuracy |
| 4 | Visual Entailment | SNLI-VE | Flickr30K | 31K | 507K | Accuracy |
| 5 | Image-Text Retrieval | COCO | COCO | 92K | 460K | Recall@1,5,10 |
| | | Flickr30K | Flickr30K | 32K | 160K | |
| 6 | RE Comprehension | RefCOCO | COCO | 20K | 142K | Accuracy |
| | | RefCOCO+ | | 20K | 142K | |
| | | RefCOCOg | | 26K | 95K | |

Table 2: Statistics on the datasets of downstream tasks.

3) **Masked Region Classification with KL-Divergence (MRC-kl)** MRC takes the most likely object class from the object detection model as the hard label (w.p. 0 or 1), which assumes the detected object class is the ground-truth label for the region. However, this may not be true, as no ground-truth label is provided for the detected region. Thus, in MRC-kl, we avoid this assumption by using soft label as supervision signal, which is the raw output from the detector (i.e., a distribution of object classes $\tilde{c}(\mathbf{v}_m^{(i)})$). MRC-kl aims to distill such knowledge into UNITER as Hinton et al. (2015), by minimizing the KL divergence between two distributions: $f_\theta(\mathbf{v_m}|\mathbf{v}_{\backslash\mathbf{m}}, \mathbf{w}) = \sum_{i=1}^{M} D_{KL}(\tilde{c}(\mathbf{v_m}^{(i)})||g_\theta(\mathbf{v_m}^{(i)}))$.

## 3.3 PRE-TRAINING DATASETS

We construct our pre-training dataset based on four existing V+L datasets: COCO (Lin et al., 2014), Visual Genome (VG) (Krishna et al., 2017), Conceptual Captions (CC) (Sharma et al., 2018), and SBU Captions (Ordonez et al., 2011). Only image and sentence pairs are used for our pre-training purpose, which makes the model framework more scalable, as additional image-sentence pairs are easy to harvest for further pre-training.

To study the effects of different datasets on pre-training, we divide the four datasets into two categories. The first one consists of image captioning data from COCO and dense captioning data from VG. We call it "In-domain" data, as most V+L tasks are built on top of these two datasets. To obtain a 'fair' data split, we merge the raw training and validation splits from COCO, and exclude all validation and test images that appear in downstream tasks. We also exclude all co-occurring Flickr30K (Plummer et al., 2015) images via URL matching, as both COCO and Flickr30K images were crawled from Flickr and may have overlaps[8]. The same rule was applied to Visual Genome as well. In this way, we obtain 5.6M image-text pairs for training and 131K image-text pairs for our internal validation, which is half the size of the dataset used in LXMERT (Tan & Bansal, 2019), due to the filtering of overlapping images and the use of image-text pairs only. We also use additional Out-of-domain data from Conceptual Captions (Sharma et al., 2018) and SBU Captions (Ordonez et al., 2011) for model training[9]. The statistics on the cleaned splits are provided in Table 1.

## 4 EXPERIMENTS

We evaluate UNITER on six V+L tasks (listed in Table 2), by transferring the pre-trained model to each target task and finetuning through end-to-end training. We report experimental results on two model sizes: UNITER-base with 12 layers and UNITER-large with 24 layers[10].

---

[8]A total of 222 images were eliminated through this process.

[9]We apply the same URL matching method, excluding 109 images from the training set.

[10]UNITER-base: L=12, H=768, A=12, Total Parameters=86M. UNITER-large: L=24, H=1024, A=16, Total Parameters=303M (L: number of stacked Transformer blocks; H: hidden activation dimension; A: number of attention heads). 882 and 3645 V100 GPU hours were used for pre-training UNITER-base and UNITER-large.

| Pre-training Data | | Pre-training Tasks | Meta-Sum | VQA | IR (Flickr) | TR (Flickr) | NLVR$^2$ | Ref-COCO+ |
|---|---|---|---|---|---|---|---|---|
| | | | | test-dev | val | val | dev | val$^d$ |
| None | 1 | None | 314.34 | 67.03 | 61.74 | 65.55 | 51.02 | 68.73 |
| Wikipedia + BookCorpus | 2 | MLM (text only) | 346.24 | 69.39 | 73.92 | 83.27 | 50.86 | 68.80 |
| In-domain (COCO+VG) | 3 | MRFR | 344.66 | 69.02 | 72.10 | 82.91 | 52.16 | 68.47 |
| | 4 | ITM | 385.29 | 70.04 | 78.93 | 89.91 | 74.08 | 72.33 |
| | 5 | MLM | 386.10 | 71.29 | 77.88 | 89.25 | 74.79 | 72.89 |
| | 6 | MLM + ITM | 393.04 | 71.55 | 81.64 | 91.12 | 75.98 | 72.75 |
| | 7 | MLM + ITM + MRC | 393.97 | 71.46 | 81.39 | 91.45 | 76.18 | 73.49 |
| | 8 | MLM + ITM + MRFR | 396.24 | 71.73 | 81.76 | 92.31 | 76.21 | 74.23 |
| | 9 | MLM + ITM + MRC-kl | 397.09 | 71.63 | 82.10 | 92.57 | 76.28 | 74.51 |
| | 10 | MLM + ITM + MRC-kl + MRFR | 399.97 | 71.92 | 83.73 | 92.87 | 76.93 | 74.52 |
| | 11 | MLM + ITM + MRC-kl + MRFR (w/o cond. mask) | 396.51 | 71.68 | 82.31 | 92.08 | 76.15 | 74.29 |
| Out-of-domain (SBU+CC) | 12 | MLM + ITM + MRC-kl + MRFR | 395.45 | 71.47 | 83.10 | 92.21 | 75.58 | 73.09 |
| In-domain + Out-of-domain | 13 | MLM + ITM + MRC-kl + MRFR | **402.50** | **72.27** | **84.68** | **93.69** | **77.14** | **74.72** |

Table 3: Evaluation on pre-training tasks and datasets using VQA, Image-Text Retrieval on Flickr30K, NLVR$^2$, and RefCOCO+ as benchmarks. All results are obtained from UNITER-base. Averages of R@1, R@5 and R@10 on Flickr30K for Image Retrieval (IR) and Text Retrieval (TR) are reported. Dark and light grey colors highlight the top and second best results across all the tasks trained with In-domain data.

## 4.1 DOWNSTREAM TASKS

In VQA, VCR and NLVR$^2$ tasks, given an input image (or a pair of images) and a natural language question (or description), the model predicts an answer (or judges the correctness of the description) based on the visual content in the image. For Visual Entailment, we evaluate on the SNLI-VE dataset. The goal is to predict whether a given image semantically entails an input sentence. Classification accuracy over three classes ("Entailment", "Neutral" and "Contradiction") is used to measure model performance. For Image-Text Retrieval, we consider two datasets (COCO and Flickr30K) and evaluate the model in two settings: Image Retrieval (IR) and Text Retrieval (TR). Referring Expression (RE) Comprehension requires the model to select the target from a set of image region proposals given the query description. Models are evaluated on both ground-truth objects and detected proposals[11] (MAttNet (Yu et al., 2018)).

For VQA, VCR, NLVR$^2$, Visual Entailment and Image-Text Retrieval, we extract the joint embedding of the input image-text pairs via a multi-layer perceptron (MLP) from the representation of the `[CLS]` token. For RE Comprehension, we use the MLP to compute the region-wise alignment scores. These MLP layers are learned during the finetuning stage. Specifically, we formulate VQA, VCR, NLVR$^2$, Visual Entailment and RE Comprehension as classification problems and minimize the cross-entropy loss over the ground-truth answers/responses. For Image-Text Retrieval, we formulate it as a ranking problem. During finetuning, we sample three pairs of image and text, one positive pair from the dataset and two negative pairs by randomly replacing its sentence/image with others. We compute the similarity scores (based on the joint embedding) for both positive and negative pairs, and maximize the margin between them through triplet loss.

## 4.2 EVALUATION ON PRE-TRAINING TASKS

We analyze the effectiveness of different pre-training settings through ablation studies over VQA, NLVR$^2$, Flickr30K and RefCOCO+ as representative V+L benchmarks. In addition to standard metrics for each benchmark (listed in Table 2) , we also use Meta-Sum (sum of all the scores across all the benchmarks) as a global metric.

Firstly, we establish two baselines: Line 1 (L1) in Table 3 indicates no pre-training is involved, and L2 shows the results from MLM initialized with pre-trained weights from Devlin et al. (2019). Although MLM trained on text only did not absorb any image information during pre-training, we see a gain of approximately +30 on Meta-Sum over L1. Hence, we use the pre-trained weights in L2 to initialize our model for the following experiments.

---

[11]The evaluation splits of RE comprehension using detected proposals are denoted as val$^d$, test$^d$, etc.

| Tasks | | SOTA | ViLBERT | VLBERT | Unicoder-VL | VisualBERT | LXMERT | UNITER BASE | UNITER LARGE |
|---|---|---|---|---|---|---|---|---|---|
| VQA | test-dev | 70.63 | 70.55 | 70.50 | - | 70.80 | 72.42 | 72.27 | **73.24** |
| | test-std | 70.90 | 70.92 | 70.83 | - | 71.00 | 72.54 | 72.46 | **73.40** |
| VCR | Q→A | 72.60 | 73.30 | 74.00 | - | 71.60 | - | 75.00 | **77.30** |
| | QA→R | 75.70 | 74.60 | 74.80 | - | 73.20 | - | 77.20 | **80.80** |
| | Q→AR | 55.00 | 54.80 | 55.50 | - | 52.40 | - | 58.20 | **62.80** |
| NLVR$^2$ | dev | 54.80 | - | - | - | 67.40 | 74.90 | 77.14 | **78.40** |
| | test-P | 53.50 | - | - | - | 67.00 | 74.50 | 77.87 | **79.50** |
| SNLI-VE | val | 71.56 | - | - | - | - | - | 78.56 | **79.28** |
| | test | 71.16 | - | - | - | - | - | 78.02 | **78.98** |
| ZS IR (Flickr) | R@1 | - | 31.86 | - | 42.40 | - | - | 62.34 | **65.82** |
| | R@5 | - | 61.12 | - | 71.80 | - | - | 85.62 | **88.88** |
| | R@10 | - | 72.80 | - | 81.50 | - | - | 91.48 | **93.52** |
| IR (Flickr) | R@1 | 48.60 | 58.20 | - | 68.30 | - | - | 71.50 | **73.66** |
| | R@5 | 77.70 | 84.90 | - | 90.30 | - | - | 91.16 | **93.06** |
| | R@10 | 85.20 | 91.52 | - | 94.60 | - | - | 95.20 | **95.98** |
| IR (COCO) | R@1 | 38.60 | - | - | 44.50 | - | - | 48.42 | **51.72** |
| | R@5 | 69.30 | - | - | 74.40 | - | - | 76.68 | **78.41** |
| | R@10 | 80.40 | - | - | 84.00 | - | - | 85.90 | **86.93** |
| ZS TR (Flickr) | R@1 | - | - | - | 61.60 | - | - | 75.10 | **77.50** |
| | R@5 | - | - | - | 84.80 | - | - | 93.70 | **96.30** |
| | R@10 | - | - | - | 90.10 | - | - | 95.50 | **98.50** |
| TR (Flickr) | R@1 | 67.90 | - | - | 82.30 | - | - | 84.70 | **88.20** |
| | R@5 | 90.30 | - | - | 95.10 | - | - | 97.10 | **98.40** |
| | R@10 | 95.80 | - | - | 97.80 | - | - | 99.00 | **99.00** |
| TR (COCO) | R@1 | 50.40 | - | - | 59.60 | - | - | 63.28 | **66.60** |
| | R@5 | 82.20 | - | - | 85.10 | - | - | 87.04 | **89.42** |
| | R@10 | 90.00 | - | - | 91.80 | - | - | 93.08 | **94.26** |
| Ref-COCO | val | 87.51 | | - | - | - | - | 91.64 | **91.84** |
| | testA | 89.02 | - | - | - | - | - | 92.26 | **92.65** |
| | testB | 87.05 | - | - | - | - | - | 90.46 | **91.19** |
| | val$^d$ | 77.48 | - | - | - | - | - | 81.24 | **81.41** |
| | testA$^d$ | 83.37 | - | - | - | - | - | 86.48 | **87.04** |
| | testB$^d$ | 70.32 | - | - | - | - | - | 73.94 | **74.17** |
| Ref-COCO+ | val | 75.38 | - | 78.44 | - | - | - | 82.84 | **84.04** |
| | testA | 80.04 | - | 81.30 | - | - | - | 85.70 | **85.87** |
| | testB | 69.30 | - | 71.18 | - | - | - | 78.11 | **78.89** |
| | val$^d$ | 68.19 | 72.34 | 71.84 | - | - | - | 74.72 | **74.94** |
| | testA$^d$ | 75.97 | 78.52 | 77.59 | - | - | - | 80.65 | **81.37** |
| | testB$^d$ | 57.52 | 62.61 | 60.57 | - | - | - | 65.15 | **65.35** |
| Ref-COCOg | val | 81.76 | - | - | - | - | - | 86.52 | **87.85** |
| | test | 81.75 | - | - | - | - | - | 86.52 | **87.73** |
| | val$^d$ | 68.22 | - | - | - | - | - | 74.31 | **74.86** |
| | test$^d$ | 69.46 | - | - | - | - | - | 74.51 | **75.77** |

Table 4: Results on downstream V+L tasks from UNITER model, compared with task-specific state-of-the-art (SOTA) and concurrent pre-trained models. ZS: Zero-Shot, IR: Image Retrieval and TR: Text Retrieval.

Secondly, we validate the effectiveness of each pre-training task through a thorough ablation study. Comparing L2 and L3, MRFR (L3) achieves better results than MLM (L2) only on NLVR$^2$. On the other hand, when pre-trained on ITM (L4) or MLM (L5) only, we observe a significant improvement across all the tasks over L1 and L2 baselines. When combining different pre-training tasks, MLM + ITM (L6) improves over single ITM (L4) or MLM (L5). When MLM, ITM and MRM are jointly trained (L7-L10), we observe consistent performance gain across all the benchmarks. Among the three variants of MRM (L7-L9), we observe that MRC-kl (L9) achieves the best performance (397.09) when combined with MLM + ITM, while MRC (L7) the worst (393.97). When combining MRC-kl and MRFR together with MLM and ITM (L10), we find that they are complimentary to each other, which leads to the highest Meta-Sum score. We use this as the optimal pre-training setting for further experiments.

Additionally, we validate the contributions of conditional masking through a comparison study. When we perform random masking on both modalities simultaneously during pre-training, i.e., w/o conditional masking (L11), we observe a decrease in Meta-Sum score (396.51) compared to that with conditional masking (399.97). This indicates that the conditional masking strategy enables the model to learn better joint image-text representations effectively.

Lastly, we study the effects of pre-training datasets. Our experiments so far have been focused on In-domain data. In this study, we pre-train our model on Out-of-domain data (Conceptual Captions

+ SBU Captions). A performance drop (395.45 in L12) from the model trained on In-domain data (COCO + Visual Genome) (399.97 in L10) shows that although Out-of-domain data contain more images, the model still benefits more from being exposed to similar downstream images during pre-training. We further pre-train our model on both In-domain and Out-of-domain data. With doubled data size, the model continues to improve (402.50 in L13).

### 4.3 RESULTS ON DOWNSTREAM TASKS

Table 4 presents the results of UNITER on all downstream tasks. Both our base and large models are pre-trained on In-domain+Out-of-domain datasets, with the optimal pre-training setting: MLM+ITM+MRC-kl+MRFR. The implementation details of each task are provided in Appendix A.2. We compare with both task-specific models and concurrent pre-trained models on each downstream task. SOTA task-specific models include: MCAN (Yu et al., 2019) for VQA, MaxEnt (Suhr et al., 2019) for NLVR$^2$, B2T2 (Alberti et al., 2019) for VCR, SCAN (Lee et al., 2018) for Image-Text Retrieval, EVE-Image (Xie et al., 2019) for SNLI-VE, and MAttNet for RE Comprehension (RefCOCO, RefCOCO+ and RefCOCOg)[12]. Concurrent pre-trained models include: ViLBERT, LXMERT, Unicoder-VL, VisualBERT and VLBERT.

Results show that our UNITER-large model achieves new state of the art across all the benchmarks. UNITER-base model also outperforms the others by a large margin across all tasks except VQA. Specifically, our UNITER-base model outperforms SOTA by approximately $+2.8\%$ for VCR on Q→AR, $+2.5\%$ for NLVR$^2$, $+7\%$ for SNLI-VE, $+4\%$ on R@1 for Image-Text Retrieval ($+15\%$ for zero-shot setting), and $+2\%$ for RE Comprehension.

Note that LXMERT pre-trains with downstream VQA (+VG+GQA) data, which may help adapt the model to VQA task. However, when evaluated on unseen tasks such as NLVR$^2$, UNITER-base achieves 3% gain over LXMERT. In addition, among all the models pre-trained on image-text pairs only, our UNITER-base outperforms the others by $>1.5\%$ on VQA.

It is also worth mentioning that both VilBERT and LXMERT observed two-stream model outperforms single-stream model, while our results show empirically that with our pre-training setting, single-stream model can achieve new state-of-the-art results, with much fewer parameters (UNITER-base: 86M, LXMERT: 183M, VilBERT: 221M)[13].

For VCR, we propose a two-stage pre-training approach: ($i$) pre-train on standard pre-training datasets; and then ($ii$) pre-train on downstream VCR dataset. Interestingly, while VLBERT and B2T2 observed that pre-training is not very helpful on VCR, we find that the second-stage pre-training can significantly boost model performance, while the first-stage pre-training still helps but with limited effects (results shown in Table 5). This indicates that the proposed two-stage approach is highly effective in our pre-trained model over new data that are unseen in pre-training datasets.

Different from other tasks, NLVR$^2$ takes two images as input. Thus, directly finetuning UNITER pre-trained with image-sentence pairs might not lead to optimal performance, as the interactions between paired images are not learned during the pre-training stage. Thus, we experimented with three modified settings on NLVR$^2$: ($i$) *Triplet*: joint embedding of images pairs and query captions; ($ii$) *Pair*: individual embedding of each image and each query caption; and ($iii$) *Pair-biattn*: a bidirectional attention is added to the *Pair* model to learn the interactions between the paired images.

Comparison results are presented in Table 6. The *Pair* setting achieves better performance than the *Triplet* setting even without cross-attention between the image pairs. We hypothesize that it is due to the fact that our UNITER is pre-trained with image-text pairs. Thus, it is difficult to finetune a pair-based pre-trained model on triplet input. The bidirectional attention mechanism in the *Pair-biattn* setting, however, compensates the lack of cross-attention between images, hence yielding the best performance with a large margin. This show that with minimal surgery on the top layer of UNITER, our pre-trained model can adapt to new tasks that are very different from pre-training tasks.

---

[12]MAttNet results are updated using the same features as the others. More details are provided in Appendix.

[13]The word embedding layer contains excessive rare words, thus excluded from the parameter counts.

| Stage I | Stage II | Q→A | QA→R | Q → AR |
|---------|----------|------|-------|--------|
| N | N | 72.44 | 73.71 | 53.52 |
| N | Y | 73.52 | 75.34 | 55.6 |
| Y | N | 72.83 | 75.25 | 54.94 |
| Y | Y | **74.56** | **77.03** | **57.76** |

Table 5: Experiments on two-stage pre-training for VCR. Results are from UNITER-base on VCR val split. Stage I and Stage II denote first-stage and second-stage pre-training.

| Setting | dev | test-P |
|---------|------|--------|
| Triplet | 72.76 | 73.55 |
| Pair | 75.37 | 75.97 |
| Pair-biattn | **77.14** | **77.87** |

Table 6: Experiments on three modified settings for NLVR$^2$. All models use pre-trained UNITER-base.

## 5 CONCLUSION

In this paper, we present UNITER, a large-scale pre-trained model providing UNiversal Image-TExt Representations for Vision-and-Language tasks. Three main pre-training tasks are proposed and evaluated through extensive ablation studies. Trained with both in-domain and out-of-domain datasets, UNITER outperforms state-of-the-art models over multiple V+L tasks by a significant margin. Future work includes studying early interaction between raw image pixels and sentence tokens, as well as developing more effective pre-training tasks.

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

# A  APPENDIX

## A.1  DATASET COLLECTION

As introduced, our full dataset is composed of four existing V+L datasets: COCO, Visual Genome, Conceptual Captions, and SBU Captions. The dataset collection is not simply combining them, as we need to make sure none of the downstream evaluation images are seen during pre-training. Among them, COCO is the most tricky one to clean, as several downstream tasks are built based on it. Figure 2 lists the splits from VQA, Image-Text Retrieval, COCO Captioning, Ref-COCO/RefCOCO+/RefCOCOg, and the bottom-up top-down (BUTD) detection (Anderson et al., 2018), all from COCO images.

As observed, the validation and test splits of different tasks are scattered across the raw COCO splits. Therefore, we exclude all those evaluation images that appeared in the downstream tasks. In addition, we also exclude all co-occurring Flickr30K images via URL matching, making sure the zero-shot image-text retrieval evaluation on Flickr is fair. The remaining images become the COCO subset within our full dataset, as shown in Figure 2 bottom row. We apply the same rules to Visual Genome, Conceptual Captions, and SBU Captions.

| | | | |
|---|---|---|---|
| MS COCO (raw) | train | val | test |
| VQA | train | train / val | test |
| Img-Txt Retrieval | train | train val test | |
| Img Captioning | train | train val test | test |
| RefCOCO(+/g) | val test train | | |
| BUTD | train | train val test | |
| UNITER | train | train val | |

Figure 2: Different data splits from downstream tasks based on COCO images. Our UNITER pre-training avoids seeing any downstream evaluation images.

## A.2  IMPLEMENTATION DETAILS

Our models are implemented based on PyTorch[14] (Paszke et al., 2017). To speed up training, we use Nvidia Apex[15] for mixed precision training. All pre-training experiments are run on Nvidia V100 GPUs (16GB VRAM; PCIe connection). Finetuning experiments are implemented on the same hardware or Titan RTX GPUs (24GB VRAM). To further speed up training, we implement dynamic sequence length to reduce padding and batch examples by number of input units (text tokens + image regions). For large pre-training experiments, we use Horovod[16] + NCCL[17] for multi-node communications (on TCP connections through ethernet) with up to 4 nodes of 4x V100 server. Gradient accumulation (Ott et al., 2018) is also applied to reduce multi-GPU communication overheads.

**Visual Question Answering (VQA)**    We follow Yu et al. (2019) to take 3129 most frequent answers as answer candidates, and assign a soft target score to each candidate based on its relevancy to the 10 human responses. To finetune on VQA dataset, we use a binary cross-entropy loss to train a multi-label classifier using batch size of 10240 input units over maximum 5K steps. We use AdamW optimizer (Loshchilov & Hutter, 2019) with a learning rate of $3e - 4$ and weight decay of 0.01. At inference time, the max-probable answer is selected as the predicted answer. For results on `test-dev` and `test-std` splits, both training and validation sets are used for training, and additional question-answer pairs from Visual Genome are used for data augmentation as in Yu et al. (2019).

---

[14]https://pytorch.org/
[15]https://github.com/NVIDIA/apex
[16]https://github.com/horovod/horovod
[17]https://github.com/NVIDIA/nccl

**Visual Commonsense Reasoning (VCR)**   VCR can be decomposed into two multiple-choice sub-tasks: question-answering task (Q → A) and answer-justification task (QA → R). In the holistic setting (Q → AR), a model needs to first choose an answer from the answer choices, then select a supporting rationale from rationale choices if the chosen answer is correct. We train our model in two settings simultaneously. When testing in the holistic setting, we first apply the model to predict an answer, then obtain the rationale from the same model based on the given question and the predicted answer. To finetune on VCR dataset, we concatenate the question (the qeustion and the ground truth answer) and each answer (rationale) choice from the four possible answer (rationale) candidates. The 'modality embedding' is extended to help distinguish question, answer and rationale. Cross-entropy loss is used to train a classifier over two classes (``right'' or ``wrong'') for each question-answer pair (question-answer-rationale triplet) with a batch size of 4096 input units over maximum 5K steps. We use AdamW optimizer with a learning rate of $1e-4$ and weight decay of 0.01.

Since the images and text in VCR dataset are very different from our pre-training dataset, we further pre-train our model on VCR, using MLM, MRFR and MRC-kl as the pre-training tasks. ITM is discarded because the text in VCR does not explicitly describe the image. The results of both pre-trainings on VCR are reported in Table 5 and discussed in the main text. In conclusion, for downstream tasks that contain new data which is very different from the pre-training datasets, second-stage pre-training helps further boost the performance.

In our implementation, the second-stage pre-training is implemented with a batch size of 4096 intput units, a learning rate of $3e-4$ and a weight decay of 0.01 over maximum 60K steps. After second-stage pre-traing, we finetune our model with a learning rate of $6e-5$ over maximum 8K steps.

**Natural Language for Visual Reasoning for Real (NLVR$^2$)**   NLVR2 is a new challenging task for visual reasoning. The goal is to determine whether a natural language statement is true about the given image pair. Here we discuss the three architecture variants of NLVR$^2$ finetuning in detail. Since UNITER only handles one image and one text input at pre-training, the 'modality embedding' is extended to help distinguish the additional image presented in the NLVR$^2$ task. For the *Triplet* setup, we concatenate the image regions and then feed into the UNITER model. An MLP transform is applied on the `[CLS]` output for binary classification. For the *Pair* setup, we treat one input example as two text-image pairs by repeating the text. The two `[CLS]` outputs from UNITER are then depth concatenated as the joint embedding for the example. Another MLP further transform this embedding for the final classification. For the *Pair-biattn* setup, the input format is the same as the Pair setup. As for the joint representation, instead of rely on only two `[CLS]` outputs, we apply a multi-head attention layer (Vaswani et al., 2017) on one sequence of joint image-text embeddings to attend to the other sequence of embeddings, and vice versa. After this 'bidirectional' attention interactions, a simple attentional pooling is applied on each output sequences and then a final concat+MLP layer transforms the cross-attended joint representation for true/false classification.

We finetune UNITER on NLVR$^2$ for 8K steps with a batch size of 10K input units. AdamW optimizer is used with learning rate of $1e-4$ and weight decay of 0.01.

**Image-Text Retrieval**   Two datasets are considered for this task: COCO and Flickr30K. COCO consists of 123K images, each accompanied with five human-written captions. We follow Karpathy & Fei-Fei (2015) to split the data into 82K/5K/5K training/validation/test images. Additional 30K images from MSCOCO validation set are also included to improve training as in Lee et al. (2018). Flickr30K dataset contains 31K images collected from the Flickr website, with five textual descriptions per image. We follow Karpathy & Fei-Fei (2015) to split the data into 30K/1K/1K training/validation/test splits. During finetuning, we sample two negative image-text pairs per positive sample from image and text sides, respectively. For COCO, we use batch size of 60 examples, learning rate of $2e-5$ and finetune our model for 20K steps. For Flickr30K, we finetune our model with a batch size of 120 examples and a learning rate of $5e-5$ over maximum 16K steps.

To obtain the final results in Table 4, we further sample hard negatives to facilitate the finetuning. For every $N$ steps, we randomly sample 128 negative images per text input and obtain a sparse scoring matrix for the whole training set. For each image, we choose the top 20 ranked negative sentences as hard negative samples. Similarly, we get 20 hard negative images for each sentence according to their scores. The hard negatives are sent to the model as additional negative samples.

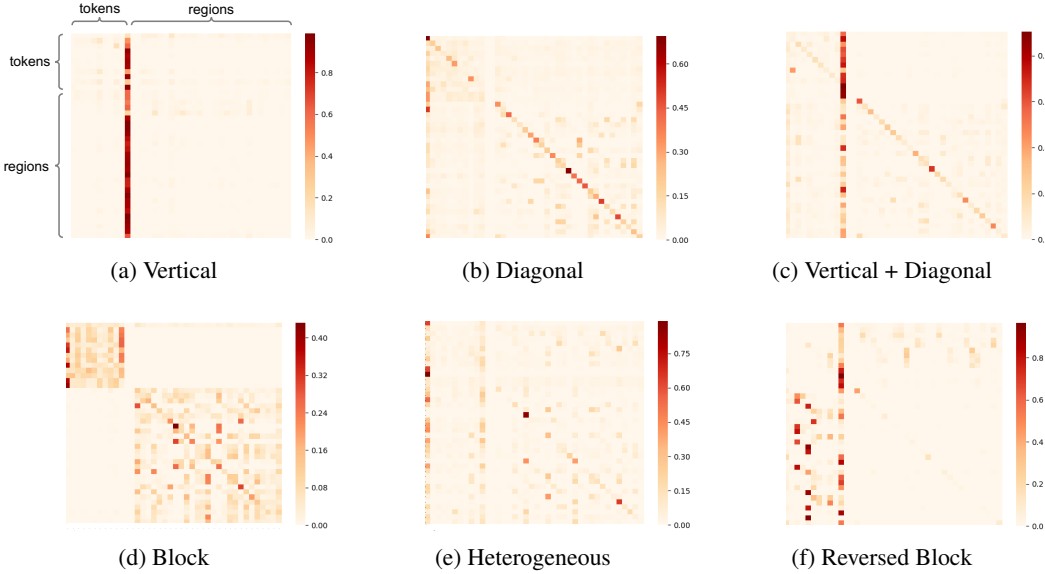

Figure 3: Attention heatmaps of UNITER

In the end, we have two randomly sampled negatives and two hard negative samples per positive sample. $N$ is set to 4000 for COCO and 2500 for Flickr30K.

**Visual Entailment (SNLI-VE)**  Visual Entailment is a task derived from Flickr30K images and Stanford Natural Language Inference (SNLI) dataset, where the goal is to determine the logical relationship between a natural language statement and an image. Similar to BERT for Natural Language Inference (NLI), we treat SNLI-VE as a three-way classification problem and apply an MLP Transform on [CLS] output. The UNITER model is finetuned using cross-entropy loss. The batch size is set to 10K input units and we use AdamW with learning rate of $8e-5$ to train for 3K steps.

**Referring Expression Comprehension**  We use three referring expression datasets: RefCOCO, RefCOCO+, and RefCOCOg for the evaluation, all collected on COCO images. To finetune UNITER on this task, we add a MLP layer on top of the region outputs from Transformer, to compute the alignment score between the query phrase/sentence and each region. Since only one object is paired with the query phrase/sentence, we apply cross-entropy loss on the normalized alignment scores. The finetuning is efficient - we train the model with a batch size of 64 examples and a learning rate of $5e-5$ for only 5 epochs, and achieve state-of-the-art performance.

Note all works including ours use off-the-shelf object detectors trained on COCO (and Visual Genome) to extract the visual features. While this does not affect other downstream tasks, it raises an issue for RE comprehension, as the val/test images of RefCOCO, RefCOCO+, and RefCOCOg are a subset of COCO's training split. Strictly, our object detector is not allowed to train with these val/test images. However, just for a "fair" comparison with concurrent works, we ignore this issue and use the same features (Anderson et al., 2018) as the others. We also update the results of MAttNet using this "contaminated" features, whose accuracy is 1.5% higher than the original one. As aforementioned, the interaction between sentence and image could start from tokens and pixels instead of the extracted features. We leave this study and RE comprehension with strictly correct features to future work.

## A.3  VISUALIZATION

Similar to Kovaleva et al. (2019), we observe several patterns in the attention maps of the UNITER model, as shown in Fig. 3. Note that different from Kovaleva et al. (2019), our attention mechanism

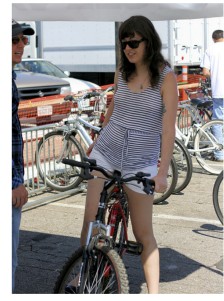
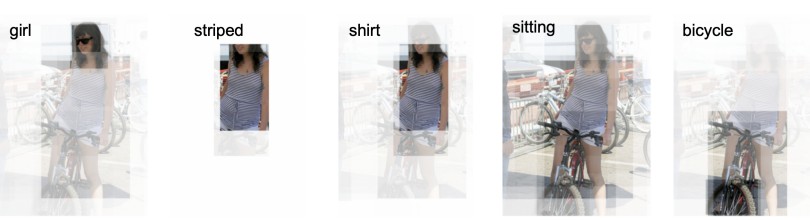

Figure 4: Text-to-image attention visualization example.

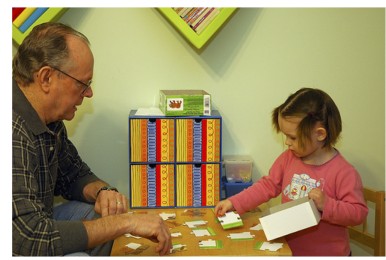
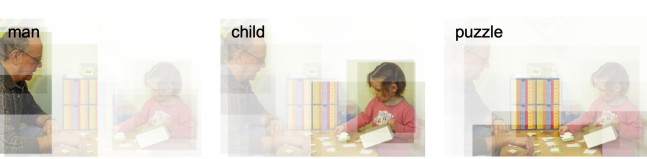

Figure 5: Text-to-image attention visualization example.

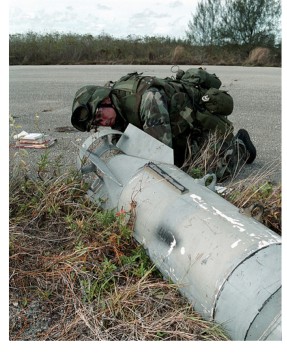
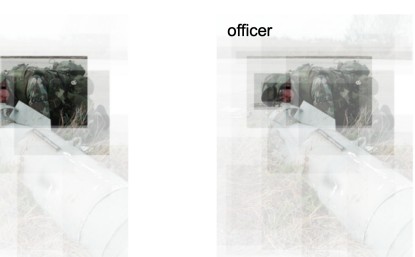
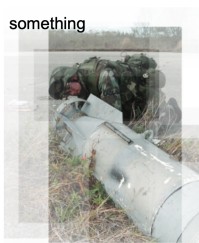

Figure 6: Text-to-image attention visualization example.

operates in both inter- and intra-modalitiy manners. For completeness, we briefly discuss each pattern here:

- *Vertical*: attention to special tokens `[CLS]` or `[SEP]`;

- *Diagonal*: attention to the token/region itself or preceding/following tokens/regions;

- *Vertical + Diagonal*: mixture of vertical and diagonal;

- *Block*: intra-modality attention, *i.e.*, textual self-attention and visual self-attention;

- *Heterogeneous*: diverse attentions that cannot be categorized and is highly dependent on actual input;

- *Reversed Block*: inter-modality attention, *i.e.*, text-to-image and image-to-text attention.

Note that *Reversed Block* (Fig. 3f) shows cross-modality alignment between tokens and regions. In Fig. 4, 5, and 6, we visualize several examples of text-to-image attention to demonstrate the local cross-modality alignment between regions and tokens.

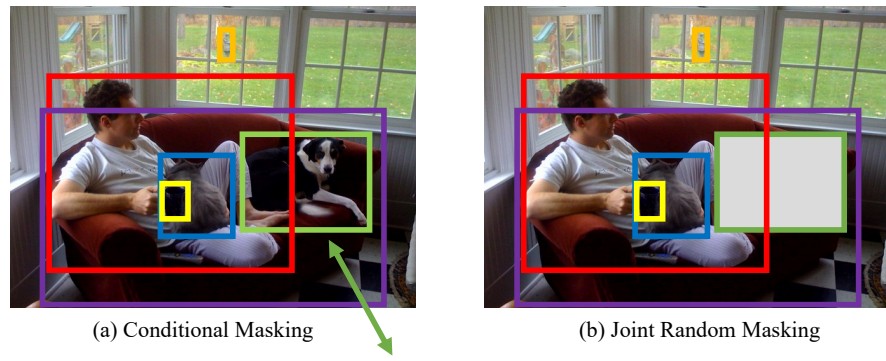

| (a) Conditional Masking | (b) Joint Random Masking |

a man with his <MASK> and cat sitting on the sofa

Figure 7: Example showing difference between conditional masking and joint random masking.

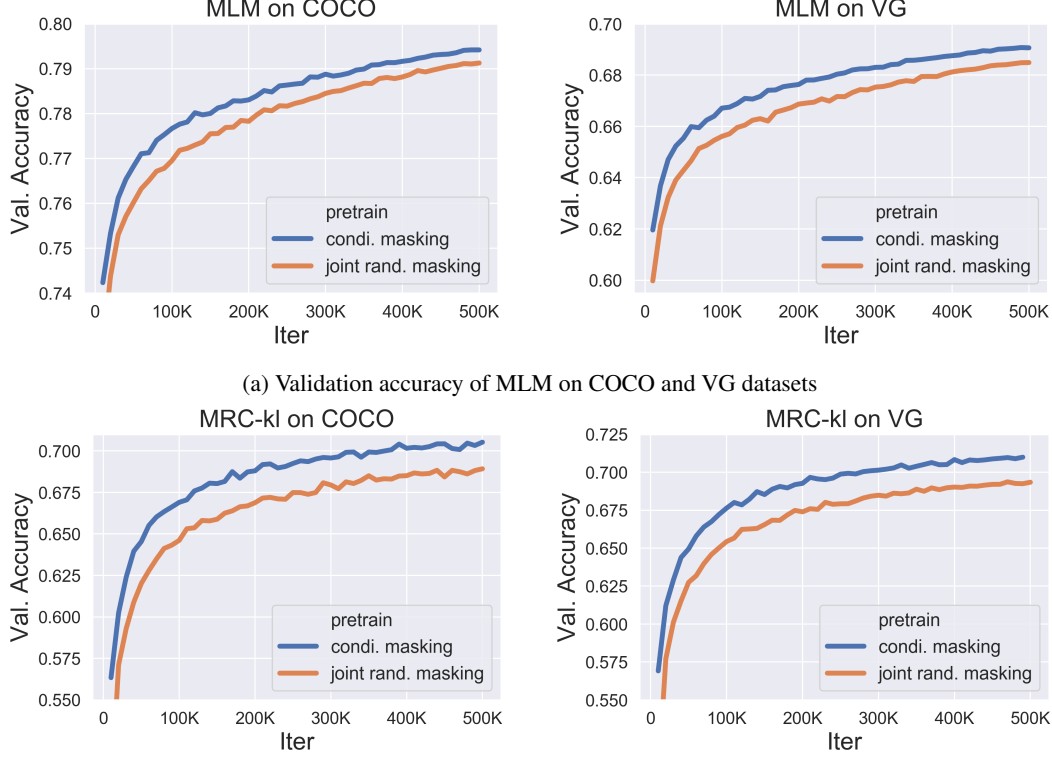

(a) Validation accuracy of MLM on COCO and VG datasets

(b) Validation accuracy of MRC-kl on COCO and VG datasets.

Figure 8: Comparison of MLM and MRC-kl validation accuracy using joint masking and our proposed conditional masking.

### A.4 CONDITIONAL MASKING VS. JOINT RANDOM MASKING

We further discuss the advantage of our proposed conditional masking over joint random masking used in (Tan & Bansal, 2019; Lu et al., 2019). Intuitively, our conditional masking learns better latent alignment of entities (regions and words) across two modalities. Fig. 7 shows an example image with "man with his dog and cat sitting on a sofa". With conditional masking, when the region of dog is masked, our model should be able to infer that the region is dog, based on the context of both surrounding regions and the full sentence (Fig. 7(a)), and vice versa. However, for the joint masking implementation, it could happen when both the region of dog and the word dog are

| Model | Q→A | QA→R | Q → AR |
|---|---|---|---|
| VLBERT-large (single) | 75.8 | 78.4 | 59.7 |
| ViLBERT (10 ensemble) | 76.4 | 78.0 | 59.8 |
| UNITER-large (single) | 77.3 | 80.8 | 62.8 |
| UNITER-large (10 ensemble) | **79.8** | **83.4** | **66.8** |

Table 7: VCR results from VLBERT (Su et al., 2019), ViLBERT (Lu et al., 2019), and UNITER.

| Model | Balanced | Unbalanced | Overall | Consistency |
|---|---|---|---|---|
| VisualBERT | 67.3 | 68.2 | 67.3 | 26.9 |
| LXMERT | 76.6 | 76.5 | 76.2 | 42.1 |
| UNITER-large | **80.0** | **81.2** | **80.4** | **50.8** |

Table 8: NLVR results on test-U split from VisualBERT (Li et al., 2019b), LXMERT (Tan & Bansal, 2019), and UNITER.

masked (Fig. 7(b)). In such case, the model has to make the prediction blindly, which might lead to mis-alignment.

To verify this intuition, we show the validation curves during pre-training of MLM and MRC-kl in Fig. 8. Each sub-figure shows a comparison between applying conditional masking and joint random masking during the pre-training of UNITER. The MLM accuracy measures how well UNITER can reconstruct the masked words, and MRC-kl accuracy[18] measures how well UNITER can classify the masked regions. In both cases, as shown in Fig. 8, our conditional masking converges faster and achieves higher final accuracy than joint random masking. In addition, Table 3 (row 10 & 11) shows our conditional masking also performs better on fine-tuned downstream tasks.

## A.5 MORE RESULTS ON VCR AND NLVR2

Following the VCR setup in Table. 5, we further construct an ensemble model using 10 UNITER-large. Table. 7 shows the comparison between VLBERT, ViLBERT and UNITER on VCR. The $Q \rightarrow AR$ accuracy of our ensemble model outperforms ViLBERT (Lu et al., 2019) ensemble by a large margin of 7.0%. Note even single UNITER-large already outperforms ViLBERT ensemble and VLBERT-large by 3.0%.

Besides, we also compare our UNITER-large with LXMERT (Tan & Bansal, 2019) and Visual-BERT (Li et al., 2019b) on an additional testing split of NLVR$^2$ in Table. 8. Our results consistently outperform the previous SOTA on all metrics[19] by a large margin of ∼4.0%.

## A.6 DIRECT COMPARISON TO VLBERT AND VILBERT

| | VQA | RefCOCO+ (det) | | |
|---|---|---|---|---|
| | test-dev | val | testA | testB |
| ViLBERT | 70.55 | 72.34 | 78.52 | 62.61 |
| VLBERT-base | 71.16 | 71.60 | 77.72 | 60.99 |
| UNITER-base | **71.22** | **72.49** | **79.36** | **63.65** |

Table 9: A direct comparison between ViLBERT (Lu et al., 2019), VLBERT (Su et al., 2019), and our UNITER, all trained on Conceptual Captions (Sharma et al., 2018) only.

To further demonstrate our idea, we conduct a direct comparison to ViLBERT (Lu et al., 2019) and VLBERT (Su et al., 2019), trained on Conceptual Captions (Sharma et al., 2018). We pre-train UNITER on Conceptual Captions only (instead of 4 datasets in 3.3) using our proposed conditional masking and the best pre-training tasks (MLM + ITM + MRC-kl + MRFR). Table. 9 shows that

---

[18]When validating on MRC-kl accuracy, we simply pick the most confident category from the predicted probability and measure its correctness.

[19]The balanced and unbalanced evaluations were introduced in Suhr & Artzi (2019).

UNITER still consistently outperforms the other models by a visible margin on VQA and Ref-COCO+.

