# OpenReview forum: "UNITER: Learning UNiversal Image-TExt Representations"
_ICLR.cc/2020/Conference — Reject_

### Official Review · AnonReviewer1 · 2019-10-23
**Official Blind Review #1**

**Rating:** 6

**Review:**

This paper presents a novel method for image-text representations called UNITER. The proposed method has been subsequently tested in many downstream tasks. A detailed ablation study helps to understand the role of each pretrained task in the proposed model.

Although the empirical results are nice, performing the intensive set of experiments on many different tasks is definitely time-consuming and needs a lot of engineering efforts, the technical contribution does not seem significant to me. The paper modifies an existing pre-training procedure by conditional masking (Section 2). I agree this is well-motivated but it has little novelty and a similar idea is there in VQA (See “Dynamic fusion with intra and inter-modality attention flow for visual question answering”). MLM and MRM are not new training procedure either, they are basically extending the BERT’s training procedure with the consideration of multiple modalities.

I have some questions for the authors:
(1) What are the advantages of using single-stream transformer over two-stream transformer (page 2). I guess it leads to fewer parameters but I don’t think this is a big problem.
(2) Some visualization of attention weights would be helpful.
Minor
•	In “m \e N^M” (equation 1), what is N and M?


**Experience Assessment:**

I have read many papers in this area.

**Review Assessment: Checking Correctness Of Derivations And Theory:**

N/A

**Review Assessment: Checking Correctness Of Experiments:**

I carefully checked the experiments.

**Review Assessment: Thoroughness In Paper Reading:**

I read the paper thoroughly.

---

> ### Author Response · Authors · 2019-11-09
> **Author Response to Reviewer #1: Comparison with DFAF**
>
>
> --------------------original question---------------------------
> The paper modifies an existing pre-training procedure by conditional masking (Section 2). I agree this is well-motivated, but it has little novelty and a similar idea is there in VQA (See “Dynamic fusion with intra and inter-modality attention flow for visual question answering”). MLM and MRM are not new training procedure either, they are basically extending the BERT’s training procedure with the consideration of multiple modalities.
> -----------------------------------------------------------------------
>
> Thank you for referring us a related work. After a thorough check of the paper, we agree with the reviewer this is also a relevant work. Nevertheless, Gao et al., (2019) may focus on novel model architecture, while we proposed a generic V+L representation via pretraining. We will cite the paper and discuss it in the related work.
> Secondly, we argue that UNITER is not trivially derived from BERT. Even for BERT, language modeling has been around for years (CBOW, Mikolov et al., 2013).
> Intuitively, mask-then-reconstruct is helpful for learning contextualization, but the key is what exact pretraining tasks are effective, especially for learning “alignment” across vision and language in our case.
> That’s why we proposed ITM, MLM, MRM (3 variants), enumerate their combinations on large-scale pretraining, and then study a diverse set of V+L downstream tasks to derive the best pretraining strategy (Table 3).
> To explain the superior performance of UNITER, we believe the conditioned MLM/MRM better learns the local alignment (token/region level) across modalities and ITM enhances the global alignment.

---

> ### Author Response · Authors · 2019-11-09
> **Author Response to Reviewer #1:  single-stream vs two-stream**
>
>
> --------------------original question---------------------------
> I have some questions for the authors:
> (1) What are the advantages of using single-stream transformer over two-stream transformer (page 2). I guess it leads to fewer parameters, but I don’t think this is a big problem.
> -----------------------------------------------------------------------
>
> Our argument is not about “single-stream being strictly better than two-stream”.  In fact, we started with a SOTA two-stream model (MCAN, Yu et al., 2019) in our preliminary experiment but found that it did not work as well as single-stream model. We therefore continued with the single-stream architecture, and focused on finding the most effective pretraining strategy, which is our main contribution. Note that both LXMERT and ViLBERT promoted two-stream models, yet we showed that single-stream model is sufficient to learn contextual representations across two modalities.
>
> Fewer parameters could potentially be an advantage, e.g., we are able to stack deeper/larger transformer layers under the same memory constraint.

---

> ### Author Response · Authors · 2019-11-09
> **Author Response to Reviewer #1: Visualization and Others**
>
>
> --------------------original question---------------------------
> (2) Some visualization of attention weights would be helpful.
> -----------------------------------------------------------------------
>
> Thank you for your suggestion. We already find some interesting patterns on the attention weights. We are working on the visualization and will update the paper before the discussion period ends.
>
> --------------------original question---------------------------
> Minor • In “m \e N^M” (equation 1), what is N and M?
> -----------------------------------------------------------------------
>
> \mathbb{N} stands for natural numbers (non-negative integers), M is the number of masked tokens, and \mathbf{m} is the set of masked indices. We will add a footnote to make this clearer.

---

### Official Review · AnonReviewer2 · 2019-10-23
**Official Blind Review #2**

**Rating:** 6

**Review:**

# 1. Summary
The authors introduce a new pre-training procedure for image-text representations. The idea is to train the model on a huge collection of different image-text datasets and the use the model for downstream tasks. The difference between the proposal wrt the concurrent work is that conditioned masking is used: (i) Masked Language Modeling (MLM) conditioned on image; (ii) Masked Region Modeling (MRM) conditioned on text; and (iii) joint Image-Text Matching (ITM).

I am on the fence for this paper given the balance between strengths and  weaknesses listed below. I am conservative here and decide for weak reject; but I am open for discussion, if the authors answer to my concerns detailed below.

Strengths:
* State-of-the-art results on several downstream vision-language tasks
* Empirical work to investigate different ways to perform conditioned masking

Weaknesses:
* Some parts of the method needs clarification (see point 2 below) to better understand the details and practical advantages of the method.
* Limited novelty: the paper is an extension of BERT to the visual domain (see point 3 below)


# 2. Clarity and Motivation
The paper reads quite well, although some points need to be improved:
* "Compared with LXMERT (Tan & Bansal, 2019) and ViLBERT (Lu et al., 2019) that use two streams (one Transformer for each modality), our UNITER model can learn joint contextualized ...", why is this an advantage? Using two streams might also lead to learning context? Maybe an example can clarify my question.
* End of Sec. 3.1 (and paragraph in Sec. 3.2): not clear how the model is training for ITM. What's the input and output? Why do you need a new symbol [CLS]?
* Sec. 3.2 ITM: "an additional special token [CLS] is fed into our model, which indicates the fused representation of both modalities" - This is not clear. Why this special token is needed? Why is not needed in the MLM and MRM?
* "The scoring function is denoted as s" -> please indicate in the main text what function you used
* MRFM and MRC are clear, however the intuition of MRC-kl is missing. Why is this needed? What does it mean in practice to minimize such divergence (provide practical example)?
* Combination of tasks (MLM + ITM + MRC-kl + MRFR) -> it is not clear how this is done in practice. Is the loss function composed (summed)? Within the mini-batch, the method randomly chooses which operation to do (e.g., MLM) for each sample? This should be clarified in the main text of the paper.


# 3. Novelty
The novelty of the paper is quite limited since it is an extension of BERT to the visual domain. The authors propose an empirical analysis of different ways to mask the visual input, however this might not be a substantial extension of previous work. In fact, recently there are many other papers (ViLBERT, VisualBERT, LXBERT, ...) working on similar topic with small differences. What it is missing in this paper is an understanding and intuition on the reasons why the conditioned masking idea should be better than the other visual masking ideas proposed in previous work.


# 4. Experimentation
The main advantage of this paper relies on the extensive experimental analysis done on many challenging datasets reaching the state of the art on several downstream tasks.
The evaluation on both pre-training tasks and downstream tasks show that the method is working well in practice.

**Experience Assessment:**

I have read many papers in this area.

**Review Assessment: Checking Correctness Of Derivations And Theory:**

N/A

**Review Assessment: Checking Correctness Of Experiments:**

I assessed the sensibility of the experiments.

**Review Assessment: Thoroughness In Paper Reading:**

I made a quick assessment of this paper.

---

> ### Author Response · Authors · 2019-11-09
> **Author Response to Reviewer #2: Comparison with LXMERT and ViLBERT**
>
>
> --------------------original question---------------------------
> * "Compared with LXMERT (Tan & Bansal, 2019) and ViLBERT (Lu et al., 2019) that use two streams (one Transformer for each modality), our UNITER model can learn joint contextualized ...", why is this an advantage? Using two streams might also lead to learning context? Maybe an example can clarify my question.
> -----------------------------------------------------------------------
>
> Our argument is not about “single-stream being strictly better than two-stream”.  In fact, we have tried a SOTA two-stream model (MCAN, Yu et al., 2019) in our preliminary experiment before the aforementioned related works are published, and found that it did not work as well as single-stream model. We therefore continued with the single-stream architecture, and focused on finding the most effective pretraining strategy, which is our main contribution. Note that both LXMERT and ViLBERT promoted two-stream models, yet we showed that single-stream model is sufficient to learn contextual representations across two modalities.

---

> ### Author Response · Authors · 2019-11-09
> **Author Response to Reviewer #2: [CLS] in ITM**
>
>
> --------------------original question---------------------------
> * End of Sec. 3.1 (and paragraph in Sec. 3.2): not clear how the model is training for ITM. What's the input and output? Why do you need a new symbol [CLS]?
>
> * Sec. 3.2 ITM: "an additional special token [CLS] is fed into our model, which indicates the fused representation of both modalities" - This is not clear. Why this special token is needed? Why is not needed in the MLM and MRM?
> ----------------------------------------------------------------------
>
>
> Thank you for the questions, we will update the paper to make it clearer.
>
> For ITM, the input is a sentence and a set of image regions and the output is a binary label (0 for negative match, and 1 for positive match). During training, we sample negative examples for each positive example by replacing the sentence/image. We extract the representation of [CLS] token as the joint representation of the input text and image, then fed into a fully connected layer to predict a score between 0 and 1. The ITM supervision is over the [CLS] token.
>
> In practice, the [CLS] token is fed into the model for all other pretraining tasks and the downstream finetuning tasks as well. However, in MLM/MRM, the goal is to reconstruct the masked token/region. Therefore, the MLM/MRM supervision is over the representation of the masked token/region.
>
> The supervision over the [CLS] token in pretraining also alleviates the input mismatch between pretraining tasks and downstream finetuning tasks, since most of the downstream tasks also regard the representation of [CLS] token as the joint representation.

---

> ### Author Response · Authors · 2019-11-09
> **Author Response to Reviewer #2: scoring function s**
>
>
> --------------------original question---------------------------
> * "The scoring function is denoted as s" -> please indicate in the main text what function you used
> ----------------------------------------------------------------------
>
> We used sigmoid function. We will make it clear in the paper. Thanks for the suggestion.

---

> ### Author Response · Authors · 2019-11-09
> **Author Response to Reviewer #2: MRC vs MRC-kl**
>
>
> --------------------original question---------------------------
> * MRFM and MRC are clear, however the intuition of MRC-kl is missing. Why is this needed? What does it mean in practice to minimize such divergence (provide practical example)?
> ----------------------------------------------------------------------
>
> In MRC, we assume the object class with the highest score to be the ground-truth label for a detected region. However, it may not be true, since no ground-truth label is provided for a detected region. In MRC-kl, we avoid making such an assumption by using a soft label instead of a hard one. This can be understood as distilling the knowledge (Hinton et al., 2015) from a pretrained object detection model to our UNITER model. Further, this hypothesis is empirically verified in our experiments (Table 1, row7&9).

---

> ### Author Response · Authors · 2019-11-09
> **Author Response to Reviewer #2: How to combine tasks**
>
>
> --------------------original question---------------------------
> * Combination of tasks (MLM + ITM + MRC-kl + MRFR) -> it is not clear how this is done in practice. Is the loss function composed (summed)? Within the mini-batch, the method randomly chooses which operation to do (e.g., MLM) for each sample? This should be clarified in the main text of the paper.
> ----------------------------------------------------------------------
>
>
> Thank you for the suggestion. We will update the paper accordingly to make this clearer. In our implementation, we randomly sample a pretraining task for each mini-batch and train on only 1 objective per SGD update following MT-DNN (Liu et al, 2019).
>
> It is also worth noting that existing implementations (LXMERT, ViLBERT) applied MLM, MRM on negatively sampled ITM pairs and summed the losses, which means 50% of the training is not correctly conditioned across modalities.

---

> ### Author Response · Authors · 2019-11-09
> **Author Response to Reviewer #2: Difference between UNITER and other concurrent works**
>
>
> --------------------original question---------------------------
> # 3. Novelty The novelty of the paper is quite limited since it is an extension of BERT to the visual domain. The authors propose an empirical analysis of different ways to mask the visual input, however this might not be a substantial extension of previous work. In fact, recently there are many other papers (ViLBERT, VisualBERT, LXBERT, ...) working on similar topic with small differences. What it is missing in this paper is an understanding and intuition on the reasons why the conditioned masking idea should be better than the other visual masking ideas proposed in previous work.
> ----------------------------------------------------------------------
>
> We think these peer works are concurrent to our UNITER. We did feel much pressure when our peer works went public, but we decided to complete all 9 downstream tasks with 13 settings (covering nearly all popular V+L tasks) to show UNITER’s better generalization ability, rather than publishing a premature work. With such extensive experiments, our work is much more convincing.
>
> It is true that all concurrent works used visual masking and language masking. However, it is not clear what the exact visual tasks are helpful for V+L self-supervised learning. First, until recently, no one knows whether MLM can be applied to image-conditioned text modeling. Second, we propose 3 variants of Masked Region Modeling (MRM) and suggest the community the most effective combination (Table 3).
>
>
> As for why our conditional masking works better than others (joint masking), we hypothesize that UNITER learns better latent alignment of entities (regions and words) across two modalities. For example, given a sentence “man with his dog on a couch” and a corresponding image as in Figure 1. For our conditional masking, when the region of “dog” is masked, our model should be able to infer that region is “dog” based on the context of both the surrounding regions and the full sentence, and vice versa. For the joint masking implementation, it could happen when both the region of “dog” and the word of “dog” are masked, then the model has to predict blindly. Therefore, joint masking might lead to miss-alignment. We show in Table 3 (row 10&11) that our conditional masking performs better than joint masking.
>
> To further demonstrate our idea, we are working on a more direct comparison with ViLBERT and other concurrent works which were trained on Conceptual Captions (CC) only. However, we would like to emphasize that large-scale data is essential for self-supervised learning. So far, only we have succeeded in pretraining on these 4 largest public datasets (10 days x 16 V100 GPUs). Also, we are working hard to resolve legal/license issue, and we will release our best pretrained model to help future V+L research.

---

> ### Author Response · Authors · 2019-11-09
> **Author Response to Reviewer #2: Experiments**
>
>
> --------------------original question---------------------------
> # 4. Experimentation The main advantage of this paper relies on the extensive experimental analysis done on many challenging datasets reaching the state of the art on several downstream tasks. The evaluation on both pre-training tasks and downstream tasks show that the method is working well in practice.
> ----------------------------------------------------------------------
>
> We appreciate the reviewer for recognizing our effort of achieving SOTA on 9 V+L tasks under 13 settings. Making so many things work well involved a fair amount of effort, but it deserves as we show UNITER has outstanding generalization ability, regardless how different these 9 downstream tasks are. We are looking forward to seeing our pre-trained model could serve as fundamental representations for future V+L research in this community.
>
> After the submission, we made another two new SOTA on VCR and NLVR2. We ensembled our VCR model and see a 4% absolute gain (66.8 vs 62.8). Note that our single large model already outperforms all other ensembled models by a large margin (62.8 vs 59.8). Besides, our NLVR2 model outperforms the others on Test-U (80.4 vs 76.2 for accuracy; 50.8 vs 42.1 for consistency). Both results will be added in the final version.

---

### Official Review · AnonReviewer3 · 2019-10-23
**Official Blind Review #3**

**Rating:** 6

**Review:**

This is an impressive paper. LIke BERT, it proposes a tranformer based approach to derive a pre-trained network for representing images and texts. The resulting pre-trained network, used in 9 different tasks, advances the SOTA on all the tasks.
The major limitation of this paper is why. Why does it happen? How this results can be achieved? What is exactly represented in this pre-trained network. Why the tasks used for pre-training build a network that is so informative?
This is really the major obscure point of this impressive paper.


**Experience Assessment:**

I have published one or two papers in this area.

**Review Assessment: Checking Correctness Of Derivations And Theory:**

I did not assess the derivations or theory.

**Review Assessment: Checking Correctness Of Experiments:**

I assessed the sensibility of the experiments.

**Review Assessment: Thoroughness In Paper Reading:**

I made a quick assessment of this paper.

---

> ### Author Response · Authors · 2019-11-09
> **Author Response to Reviewer #3**
>
> Thank you for your insightful comments. Our assumption is that UNITER learns contextualized joint representation of both modalities. In Section 3.1, we proposed conditional masking that allows the model to learn informative representation of one modality conditioned on the other. Note that the conditional masking happens in both modalities. Therefore, the representation is aware of both visual and textual information. The reconstruction of masked tokens/regions can be viewed as learning local alignment across modalities. Furthermore, when combined with ITM pretraining, the global alignment between both modalities is encouraged. We also show that every pretraining task contributes to the final performance gain. To better address the reviewers’ concern, we are working on attention visualization, and will update the paper before the discussion period ends.

---

### Author Response · Authors · 2019-11-15
**General Response to All Reviewers**

We thank all reviewers for your reviews. We have updated the paper and the changes are in blue for easier reference. To summarize, we have added:
  1) visualization of attention and qualitative examples;
  2) additional analysis on conditional masking vs. joint random masking;
  3) more recent SOTA on VCR and NLVR2;
  4) additional experiments on Conceptual-Caption-only pre-training;
  5) some revisions suggested by the reviewers.

---

### Decision · Program_Chairs · 2019-12-19

**Decision:**

Reject

**Comment:**

This submission proposes an approach to pre-train general-purpose image and text representations that can be effective on target tasks requiring embeddings for both modes. The authors propose several pre-training tasks beyond masked language modelling that are more suitable for the cross-modal context being addressed, and also investigate which dataset/pretraining task combinations are effective for given target tasks.

All reviewers agree that the empirical results that were achieved were impressive.

Shared points of concern were:
- the novelty of the proposed pre-training schemes.
- the lack of insight into the results that were obtained.

These concerns were insufficiently addressed after the discussion period, particularly the limited novelty. Given the remaining concerns and the number of strong submissions to ICLR, this submission, while promising, does not meet the bar for acceptance.